# Virological and Clinical Outcome of DAA Containing Regimens in a Cohort of Patients in Calabria Region (Southern Italy)

**DOI:** 10.3390/medicina56030101

**Published:** 2020-02-28

**Authors:** Vincenzo Scaglione, Maria Mazzitelli, Chiara Costa, Vincenzo Pisani, Giuseppe Greco, Francesca Serapide, Rosaria Lionello, Valentina La Gamba, Nadia Marascio, Enrico Maria Trecarichi, Carlo Torti

**Affiliations:** 1Infectious and Tropical Disease Unit, Department of Medical and Surgical Sciences, “Magna Graecia” University of Catanzaro, 88100 Catanzaro, Italy; vincenzo.scaglione91@gmail.com (V.S.); grecopep@gmail.com (G.G.); francescaserapide@gmail.com (F.S.); roli88@tiscali.it (R.L.); valentina.lagamba@alice.it (V.L.G.); em.trecarichi@unicz.it (E.M.T.); torti@unicz.it (C.T.); 2Unit of Infectious and Tropical Diseases, “Mater Domini” Teaching Hospital, 88100 Catanzaro, Italy; ch.costa@tin.it (C.C.); vin.pisani@virgilio.it (V.P.); 3Clinical Microbiology Unit, Department of Heath Sciences, “Magna Graecia” University of Catanzaro, 88100 Catanzaro, Italy; nadiamarascio@gmail.com

**Keywords:** HCV, DAA, real-life, HCC, outcome

## Abstract

*Background and objectives:* In Italy, Hepatitis C Virus (HCV) infections are most prevalent in people older than 50 years of age, who often experience multi-morbidities, take co-medications, and have a long history of liver disease. These characteristics could potentially affect tolerability of HCV treatments and adherence in this subgroup. After achievement of sustained virological response (SVR), retention into care is very important both to detect the onset of possible complications and prevent further infections. In this study, SVR rates and retention into care of patients treated with directly acting antivirals (DAAs) of a single-center cohort in Southern Italy were evaluated. *Materials and Methods:* Patients treated with directly acting antivirals from 2014 to 2018 were included. Patients were stratified by age (i.e., <65 vs. ≥65 years) and by cirrhosis presence (i.e., liver stiffness >14.6 KPa or clinical/ultrasound cirrhosis vs. absence of these criteria). Primary outcome was availability of SVR at Weeks 12–24 after the end of treatment. Inter- and intra-group comparisons were performed along the follow-up for significant laboratory parameters. *Results:* In total, 212 patients were treated; 184 (87%) obtained SVR after the first treatment course and 4 patients after retreatment. Twenty-two (10.4%) patients were lost to follow-up before assessment of SVR, and two patients died before the end of treatment for liver decompensation. Considering only the first treatment episode, per protocol analysis (i.e., excluding patients lost to follow-up) showed the following rates of SVR: 97% (overall), 97% (older age group), 96% (age group <65 years), 94% (cirrhotics), and 100% (non-cirrhotics). By contrast, at the intention to treat analysis (i.e., patients lost were computed as failures), SVR percentages were significantly lower for patients <65 years of age (80%) and for non-cirrhotics (85%). *Conclusions:* High rates of SVR were obtained. However, younger patients and those without cirrhosis displayed an apparent high risk of being lost to follow-up. This may have important implications: since those who are lost may transmit HCV in case SVR is not achieved, these subpopulations should receive appropriate counselling during treatment.

## 1. Introduction

Hepatitis C Virus (HCV) infection is one of the major causes of liver disease, and individuals with chronic infection are at high risk both of liver- and non-liver-related morbidity and mortality [1].

In the last few years, Directly Acting Antivirals (DAAs) drugs revolutionized the therapeutic approach. Therefore, an increasing number of patients became eligible for antiviral treatment, including those who did not tolerate interferon (IFN)-based treatment or for whom interferon was contra-indicated [2,3,4,5].

In Italy, the majority of HCV infections are in people older than 50 years of age [6]. Elderly people often experience multi-morbidities (either due to aging or to HCV itself), take comedications, and most likely also have a long history of hepatic disease, even for the risk factors associated with the infection in Italy [2]. All these characteristics (advanced age, comorbidities, and high burden of comedications) could potentially affect tolerability of HCV treatment and adherence in this subgroup [7]. Moreover, some possible side effects and biomarker alterations, as well as increased risk of liver cancer, have been reported by several authors, especially in elderly people [8,9,10,11,12,13]. In addition, the presence of some side effects was in particular reported among those who had concurrent comorbidities (such as liver cirrhosis, cardiac failure, and advanced age) [14,15]. Most people who have been treated with DAAs in Italy over the last years are older than 65 years of age.

However, the number of elderly patients enrolled in clinical trials for anti-HCV medications approval or during drug registration procedures is generally quite small [16]. Thus, reaching meaningful conclusions or generalizing results for this population from clinical trial results is difficult. This issue may raise some concerns about tolerability and adherence to treatment in subjects with advanced age and several comorbidities.

In this study, SVR rates and retention into-care of patients treated with DAAs of a single-center cohort in Southern Italy were evaluated, reporting a real-life experience on HCV treatment. In particular, outcome (defined as achievement of Sustained Virological Response, SVR) and retention into care were evaluated. Patients were stratified by age (i.e., <65, young group vs. ≥65 years, elderly group) and absence/presence of liver cirrhosis.

## 2. Materials and Methods

A retrospective study was conducted in patients treated for chronic HCV infection from January 2014 to December 2018 at Infectious and Tropical Disease Unit of Mater Domini teaching hospital (Southern Italy). This analysis was performed as part of the SINERGIE study protocol [17]. All patients gave their informed consent for participation in this study. The study was conducted in accordance with the Declaration of Helsinki, and the study protocol was approved by the Ethics Committee of Calabria Region (project identification code #2012.58.E; 19 June 2013) [17].

Data were collected from clinical records. The following data were collected: age, list of comorbidities, liver stiffness at baseline measured by transient elastography (KPa), virological data (HCV RNA viral load at baseline and HCV genotype), complete blood count, biochemical data (GOT, GPT, γGT, creatinine, total bilirubin, and albumin levels), and alpha-fetoprotein levels. Liver fibrosis was estimated by transient elastography, Fibrosis-4 (FIB-4) score, or aspartate transaminase (AST) to platelet ratio index (APRI). According to transient elastography, patients were considered as cirrhotic when estimated liver stiffness was ≥14.5 kPa [18]; cirrhosis was also defined by clinical or ultrasound signs. Similarly, due to unavailability of steatosis diagnostic tools, steatosis was considered when reported by report of abdominal ultrasound. For FIB-4 calculation, the following formula was applied: $FIB-4=\frac{\left(age\times AST\right)}{\left(PLT\;count\times \sqrt{ALT}\right)}$, in which AST and ALT were measured as IU/L, platelets as number × 10^6^/μL, and age was reported in years [19].

For APRI calculation, the following formula was used: $APRI=\frac{\frac{\left(AST\right)}{\left(AST\;upper\;limit\;of\;normal\right)}}{\left(PLT\;count\right)}\times 100$, in which AST was measured as IU/L, platelets as number × 10^6^/μL, and AST upper normal limit was fixed at 41 IU/L; if APRI value was ≤0.5 or ≥1.5, cirrhosis status was excluded or included. If FIB-4 value was ≤1.45 or ≥3.25 cirrhosis status was excluded or included [20].

The primary outcome measure was documented achievement of SVR12 and being alive after the first DAA course evaluated with per protocol (PP) analysis. Another analysis was conducted by considering as denominator all people who started treatment (intention to treat, ITT analysis).

Analysis was conducted to gain insights on the possible impact of loss to follow-up on the rate of SVR both in the overall population, and in subgroups (elderly group, i.e., ≥65 years, and young group, i.e., <65 years). Retention into care was defined as proportion of people who attended clinical appointments and performed abdominal ultrasound (as for clinical indication) after the end of treatment (12 weeks after and 24 weeks after). Patients for whom HCV RNA detection was not available at 12 weeks after the EOT or onwards were considered as lost to follow-up, and as potential treatment failure, as well as those who failed treatment or died. After data collection, patients for whom data regarding SVR were not available were contacted by phone in order to collect further information on follow-up (i.e., availability of HCV RNA to assess SVR).

Results are reported as mean, standard deviation (SD), and percentages, as appropriate. Differences among proportions were evaluated by Chi-square test, whereas Student test was used for continuous variables. Logistic regression model was implemented to explore any possible factors associated with the probability to be lost to follow-up. A *p*-value lower than 0.05 was assumed as level of significance.

## 3. Results

### 3.1. Patient Characteristics and Differences

Two hundred twelve patients were included in this analysis; 120 (56.6%) were males and 92 (43.4%) females. Mean age was 65 years (SD: 13). The most common HCV genotype was 1b (132/212, 62.3%). The most prevalent comorbidities were hypertension (123/212, 58%), diabetes (46/212, 21.7%), hypothyroidism (21/212, 9.9%), and dyslipidemia (21/212, 9.9%). History of malignancies was reported in 14.2% patients (30/212), of whom four were Hepato-Cellular Carcinomas (HCC). As for liver disease, 111/212 patients (52.4%) had liver cirrhosis and 65 (30.7%) presented liver steatosis at ultrasound (“bright liver as for fatty liver infiltration”). Three patients (1.4%) also presented cryoglobulinemia before starting treatment and six (2.8%) were living with HIV. Seventy-six patients (35.8%) had experienced failure to IFN-based regimens.

Table 1 depicts socio-demographic, virological characteristics, co-morbidities, blood test results, and liver stiffness of HCV-infected patients stratified by age and presence of cirrhosis, according to age and liver disease stage.

Stratifying patients by age (i.e., ≥65 vs. <65 years), comorbidities were more common in the elderly group, as expected. In particular, statistically significant differences were observed for the prevalence of hypertension (90/123 vs. 33/89, *p* < 0.001), diabetes mellitus (33/123 vs. 13/89, *p* = 0.03), cancer (25/123 vs. 5/89, *p* = 0.002), and cirrhosis (75/123 vs. 36/89, *p* = 0.003).

Prescribed regimens are illustrated in Table 2. Most patients received sofosbuvir-based regimens (185/212, 87.3%). Overall, 37 (17.5%) patients received also ribavirin.

### 3.2. Rates of SVR and Possible Predictors

Figure 1 shows rates of SVR after DAA therapy in the sample studied. In particular, it shows results by the two different analysis performed (PP and ITT). Results of the first kind of analysis (PP) are depicted by the first column for each patient sub-group, while the second column describes the outcome of the ITT analysis.

At the ITT, SVR was achieved in 184/212 (87%) patients. For 165/184 (89.7%) patients, SVR was determined 12 weeks after the EOT, while SVR was determined 24 weeks after the EOT in only 19/184 (10.3%) patients with missed HCV RNA test at Week 12.

### 3.3. Outcome of Patients with Available Data

Among 184 patients who were retained into care and were alive, 12 (6.5%) developed HCC, which was detected within one year after the EOT. All patients who developed HCC were cirrhotic and older than 65 years of age. Twenty-eight out of 212 (13.2%) patients did not achieve SVR. Out of these 28 patients, 22 (78.6%) were lost to follow-up. Four patients had a virological failure requiring a second course of DAAs, and two patients died during therapy due to decompensating cirrhosis.

### 3.4. Predictors of Retention into Care Until SVR

At univariate analysis, variables associated with the risk of being lost to follow-up were: male gender (OR: 2.57, 95%CI: 0.98–7.50, *p* = 0.03), HCV genotype 3 (OR: 4.78, 95%CI: 1.12–18.08, *p* < 0.01), and history of HCC (OR: 7.00, 95%CI: 0.48–98.85, *p* = 0.03). By contrast, age ≥65 years (OR: 0.35, 95%CI: 0.14–0.85, *p* = 0.01) and genotypes 1b (OR: 0.28, 95%CI: 0.11–0.69, *p* < 0.01) were negatively associated with being lost to follow-up. When multivariable model was performed, protective factors against lost to follow-up risk were confirmed to be: age ≥65 years (OR: 0.38, 95%CI: 0.16–0.89, *p* = 0.02), and HCV genotype 1b (OR: 0.28, 95%CI: 0.12–0.66, *p* < 0.01). By contrast, history of HCC (OR: 10.2, 95%CI: 1.1–94.5, *p* = 0.04) was confirmed to be negatively associated with the outcome.

## 4. Discussion

In this cohort, SVR rate approximated that reported by clinical trials on DAA treatment [21]. This confirms data obtained in several cohorts from real clinical practice [22]. However, SVR achievement as ascertained by patient clinical records was suboptimal at an analysis where lack of information on SVR was considered as failure (87%). Even though patients may have responded well, despite lack of information on SVR in their charts, this result indicates that strength of follow-up should be optimized. There are already possible solutions to improve care delivery, resulting in high SVR rates for the sake of HCV treatment expansion and elimination [23].

It should be taken into account that regimens available in 2014 (such as those containing ribavirin) are suboptimal according to the current standard of care. Nevertheless, it is unlikely that poor tolerability of regimens is the explanation for the loss to follow-up. Indeed, all the combinations were well tolerated in this cohort, even in patients with multiple comorbidities and advanced age. Some authors have found that geriatric patients affected by chronic HCV infection can be safely treated with DAAs with similar effectiveness compared to younger adults [16]. However, subgroup analysis which compare patients by age are not very numerous, especially as far as real clinical practice is concerned [16].

What are the implications of these results? On the one hand, discontinuation of follow-up may lead to underdiagnosis of relevant clinical complications that could be prevented. This is likely to be a problem mostly in elderly patients, with advanced liver fibrosis or multiple comorbidities, especially if SVR is not achieved. However, in the elderly group, a lower rate of loss to follow-up was observed in the present study. Indeed, thanks to a proactive strategy of surveillance, 12 cases of liver cancer were diagnosed early after the EOT (i.e., within one year). It is possible that adherence to care process was better in elderly people because their engagement into care was more consolidated over time. Unfortunately, treatment of patients with decompensated liver cirrhosis is still an issue and remains a challenge even over the long term [24]. Two patients did not complete treatment and died for liver decompensation. In this regard, although liver function may improve after SVR in patients with advanced liver disease, SVR may be reduced in these patients, and some of them may even die early after starting treatment due to liver decompensation [25,26]. This could raise concerns about the futility of treatment at least in some patients and with the older regimens.

On the other hand, in younger patients, risk of discontinuation of follow-up may be greater (as shown by multivariate analysis). In these patients, the main risk is likely to be continuing HCV transmission if SVR is not obtained and risk behaviors (i.e., intravenous drug use) are still ongoing. Moreover, patients with ongoing intravenous drug use should receive most importantly a follow-up for reinfection assessment.

For all these reasons, each sub-population should receive a specific and targeted counselling before starting and during treatment on the importance of being adequately involved in the care process with different perspectives. The main focus should be oriented in prevention of clinical complications in the elderly, and in guarantying the assessment of SVR in young patients, in order to be sure that the risk of transmission is minimized.

Since patients are carefully selected in randomized clinical trials in order to provide good adherence to study interventions, real-life data are important in order to obtain information regarding adherence to treatment. For this reason, the present results could contribute to increase literature data. However, this study is somewhat limited by the low number of patients, the possible missing data, the retrospective nature of the study, and the lack of interventions aimed at improving treatment adherence. Importantly, it would have been useful to evaluate the actual risk of HCV transmission, since only patients with ongoing risky behaviors remain with a transmission risk, if treatment fails. Lastly, it should be considered that patients without SVR assessment most probably had similar SVR rates to those who had been tested for HCV RNA after treatment. Thus, only a very small proportion of the lost to follow-up patients would have remained HCV RNA carriers. In conclusion, the present study provides a worst-case scenario of risk of failure and consequent transmission impairing the outcome of eradication strategies, while results of the PP analysis should be interpreted as a best-case scenario informing in the real-life clinical practice.

## 5. Conclusions

Targeted populations (i.e., young people), who have an increased risk to be lost to follow-up, deserve to receive an appropriate counselling on the importance to adhere to the plan of care. In addition, surveillance for possible complications should be proactively performed, especially in patients who present an advanced liver disease before starting treatment. Importantly, most of the older patients and those with cirrhosis remained in follow-up care, facilitating early detection of possible complications such as HCC. For these patients, early HCC diagnosis remains determinant.

## Figures and Tables

**Figure 1 medicina-56-00101-f001:**
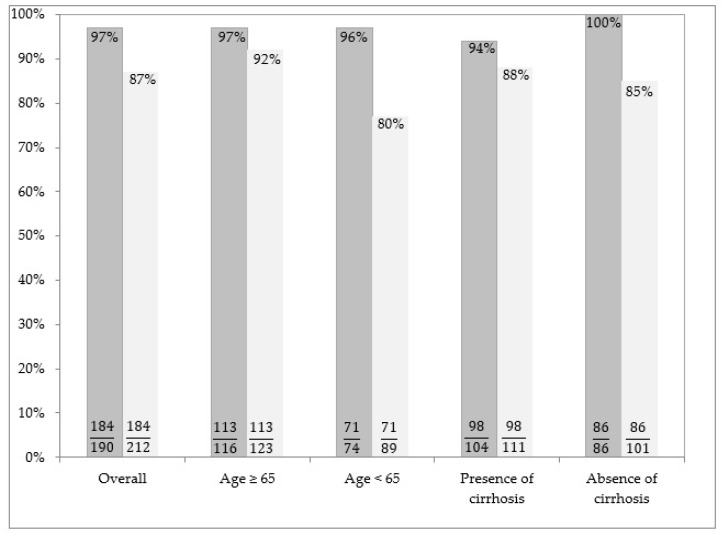
Rates of SVR after DAA therapy in the sample studied. The proportions of patients who obtained SVR among those with available data (i.e., PP analysis) are illustrated in the left columns for each patient subgroup. The proportions of patients who obtained SVR among all those who were prescribed treatment (i.e., ITT analysis) are illustrated in the right columns for each patient subgroup. In this analysis, patients with missing data are considered as failure.

**Table 1 medicina-56-00101-t001:** Qualitative and quantitative factors among HCV-infected patients treated with DAA.

Qualitative Factors							
Patients Characteristics		n = 212 (%)					
		Patients Aged ≥ 65 (n = 123)	Patients Aged < 65 (n = 89)	*p* Value	Presence of Cirrhosis (n = 111)	Absence of Cirrhosis (n = 101)	*p* Value
Gender	Male	64 (30.2)	56 (26.4)	0.11	62 (55.9)	58 (57.4)	0.817
	Female	59 (27.8)	33 (15.6)		49 (44.1)	43 (42.6)	
Genotype	1a	6 (2.8)	7 (3.3)	0.37	7 (6.3)	6 (5.9)	0.911
	1b	81 (38.2)	51 (24.1)	0.20	71 (64.0)	61 (60.5)	0.592
	2	32 (15.1)	12 (5.7)	0.26	24 (21.6)	20 (19.8)	0.744
	3	1 (0.4)	12 (5.7)	<0.001	6 (5.4)	7 (6.9)	0.643
	4	3 (1.4)	7 (3.3)	0.06	3 (2.7)	7 (6.9)	0.147
Hypertension		90 (42.4)	33 (15.6)	<0.001	71 (64.0)	52 (51.4)	0.066
Hepatic steatosis		38 (17.9)	27 (12.7)	0.93	29 (26.1)	36 (35.6)	0.133
Dyslipidaemia		16 (7.5)	5 (2.4)	0.07	13 (11.7)	8 (7.9)	0.356
Diabetes mellitus		33 (15.6)	13 (6.1)	0.03	32 (28.8)	14 (13.9)	0.008
Thyroid disease		12 (5.7)	9 (4.2)	0.93	14 (12.6)	7 (6.9)	0.166
Cancer		25 (11.8)	5 (2.4)	0.002	17 (15.3)	13 (12.9)	0.610
HCC		2 (0.9)	2 (0.9)	0.74	4 (3.6)	0	0.054
Cirrhosis		75 (35.4)	36 (17.0)	0.003	N/A	N/A	
Previous treatments		42 (19.8)	34 (16.0)	0.54	45 (40.5)	31 (30.7)	0.135
Ribavirin presence		25 (11.8)	12 (5.7)	0.19	26 (23.4)	11 (10.9)	0.016
HIV coinfection		6 (2.8)	0	0.003	1 (0.9)	5 (4.9)	0.070
Lack of assessment of SVR		10 (4.7)	18 (8.5)	0.01	13 (11.7)	15 (14.9)	0.500
**Quantitative Factors**							
		**Mean (SD)**		***p* Value**	**Mean (SD)**		***p* Value**
Stiffness (KPa)		14.5 (10.9)	12.6 (10.5)	0.31	18.2 (12.7)	8.6 (4.0)	<0.001
AST (U/L)		57.8 (68.0)	50.2 (37.8)	0.31	62.3 (51.5)	42.3 (62.2)	0.045
ALT (U/L)		61.5 (75.5)	57.8 (44.9)	0.65	67.9 (71.3)	51.2 (54.8)	0.057
γGT (U/L)		59.1 (56.3)	63.2 (48.3)	0.58	68.6 (48.0)	52.0 (57.1)	0.027
Hb (g/dL)		13.4 (1.7)	14.6 (1.7)	< 0.01	13.6 (1.7)	14.1 (1.9)	0.041
RBC (cells/µL)		4,542,231 (604,189)	4,920,920 (713,717)	< 0.01	4,607,009	4,806,869	0.035
WBC (cells/µL)		5802 (2017)	7093 (5326)	0.03	5717 (1902)	7044 (5105)	0.016
PLT (cells/µL)		17,8750 (82,743)	19,0427 (72,127)	0.28	153,754 (78,302)	21,7291 (63,952)	<0.001
Creatinine (mg/dL)		0.90 (0.30)	0.84 (0.27)	0.19	0.84 (0.25)	0.91 (0.32)	0.091
Bilirubin (mg/dL)		0.84 (0.69)	0.71 (0.52)	0.15	0.90 (0.71)	0.65 (0.49)	0.006
Albumin (mg/dL)		4.08 (0.59)	5.11 (5.57)	0.19	4.6 (4.6)	4.3 (0.4)	0.506
αFP (ng/mL)		7.3 (8.7)	8.8 (11.5)	0.41	10.8 (12.5)	4.8 (3.9)	<0.001
APRI score		1.11 (1.44)	0.79 (0.76)	0.04	1.32 (1.23)	0.59 (1.05)	<0.001
FIB-4 score		3.96 (3.55)	2.36 (1.94)	<0.001	4.5 (3.5)	1.9 (1.6)	<0.001
HCV-RNA (IU/mL)		2,815,673 (3,378,195)	2,896,647 (344,205)	0.86	3,270,946 (4,124,933)	2,386,677 (2,293,359)	0.052

**Table 2 medicina-56-00101-t002:** DAA regimens prescribed among HCV-infected patients.

DAA Regimen	n (%)
Sofosbuvir/velpatasvir	50 (23.6)
Sofosbuvir/velpatasvir + ribavirin	1 (0.4)
Sofosbuvir/ledipasvir	47 (22.2)
Sofosbuvir/ledipasvir + ribavirin	9 (4.3)
Sofosbuvir/daclatasvir	26 (12.3)
Sofosbuvir/daclatasvir + ribavirin	7 (3.3)
Sofosbuvir/simeprevir	25 (11.8)
Sofosbuvir/simeprevir + ribavirin	8 (3.8)
Elbasvir/grazoprevir	25 (11.8)
Sofosbuvir + ribavirin	12 (5.7)
Simeprevir/daclatasvir	1 (0.4)
Paritaprevir/ritonavir/ombitasvir/dasabuvir	1 (0.4)

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
