# Peer review of "Virological and Clinical Outcome of DAA Containing Regimens in a Cohort of Patients in Calabria Region (Southern Italy)"

_medicina, 2020, doi:10.3390/medicina56030101_

Round 1
Reviewer 1 Report
The Authors of the paper conducted a retrospective study among patients who had been treated for chronic HCV infection at Infectious and Tropical Disease Unit in order to assess virological and clinical outcomes of DAAs. This paper is a significant contribution to the scientific discussion about outcome of DAAs.
I suggest the revisions:
1) I suggest using logistic regression in data analysis
2) Include the paper: Koren DE, Zuckerman A, Teply R, Nabulsi NA, Lee TA, Martin MT. Expanding Hepatitis C Virus Care and Cure: National Experience Using a Clinical Pharmacist-Driven Model. Open Forum Infect Dis. 2019;6(7). pii: ofz316.
Author Response
Reviewer #1
The Authors of the paper conducted a retrospective study among patients who had been treated for chronic HCV infection at Infectious and Tropical Disease Unit in order to assess virological and clinical outcomes of DAAs. This paper is a significant contribution to the scientific discussion about outcome of DAAs.
Thank you for the high consideration of our work. We have addressed all concerns you have raised.
1) I suggest using logistic regression in data analysis
Please see lines 112-113 (method section). Logistic regression was performed.
2) Include the paper: Koren DE, Zuckerman A, Teply R, Nabulsi NA, Lee TA, Martin MT. Expanding Hepatitis C Virus Care and Cure: National Experience Using a Clinical Pharmacist-Driven Model. Open Forum Infect Dis. 2019;6(7). pii: ofz316.
We added the suggested reference (#23): “There are already possible solutions to improve care delivery, resulting in high SVR rates for the sake of HCV treatment expansion and elimination”. Please see lines 184-185.

Reviewer 2 Report
The authors present a retrospective single-center study analysing outcome in retention in care after SVR among their collective of HCV-DAA treated patients. In addition co-morbidities and their impact on retention in care is assessed.
The manuscript is well structured but it needs revision for language. Unfortunately the authors do not present potential transmission routes for their collective. This information would be relevant for the discussion and conclusion, as only patients with ongoing risk behaviour remain with a transmission risk if treatment fails.
It should be discussed that patients without SVR assessment most probably have similar SVR rates as those who have been HCV RNA tested after treatment. So only a very small proportion of the lost to follow-up patients would remain HCV RNA carriers.
Patients with cirrhosis are in need of HCC screening post-SVR. In the study most of the older patients and those with cirrhosis remain in follow-up care. This is a key finding and should be stressed more in the manuscript and be the main conclusion.
The terms intention-to-treat vs per-protocoll analysis should be used throughout the manuscript.
The authors assume that age, comorbidity and comedication could impact adherence. Please reference.
I propose not to use the first person plural form, this is unusual for scientific manuscripts.
Please provide the total time of observation for the 12 detected HCC cases.
It remains unclear whether genotype 3 is pos. or neg risk factor for retention in follow-up care.
Discussion: please do not use the term "decompensated patients". Better use "patients with decompensated liver cirrhosis.
Discussion: patient with intravenous drug use would most importantly use a follow-up for re-infection assessment. Please discuss.
Line 189: risk of HCC and progression of liver disease is only a topic for Non-SVR patients, please add this information
Author Response
Reviewer #2
The authors present a retrospective single-center study analysing outcome in retention in care after SVR among their collective of HCV-DAA treated patients. In addition co-morbidities and their impact on retention in care is assessed.
Thank you for the high consideration of our work. We have addressed all concerns raised.
1) The manuscript is well structured but it needs revision for language.
An extensive revision of English language was performed.
2) Unfortunately the authors do not present potential transmission routes for their collective. This information would be relevant for the discussion and conclusion, as only patients with ongoing risk behaviour remain with a transmission risk if treatment fails.
Thank you for this suggestion. We added the following sentence:“In these patients the main risk is likely to be continuing HCV transmission if SVR is not obtained and risk behaviours (i.e., intravenous drug use) are still ongoing. Therefore, patients with ongoing intravenous drug use should receive most importantly a follow-up for reinfection assessment” and “Importantly, it would have been useful to evaluate the actual risk of HCV transmission, since only patients with ongoing risky behaviours remain with a transmission risk, if treatment fails”. Please see lines 208-211 and 222-224.
3) It should be discussed that patients without SVR assessment most probably have similar SVR rates as those who have been HCV RNA tested after treatment. So only a very small proportion of the lost to follow-up patients would remain HCV RNA carriers.
Thank you for your suggestion. We added the following sentence: “Lastly, it should be considered that patients without SVR assessment most probably had similar SVR rates to those who had been tested for HCV RNA after treatment. So, only a very small proportion of the lost to follow-up patients would have remained HCV RNA carriers. In conclusion, the present study provides a worst-case scenario of risk of failure and consequent transmission, impairing the outcome of eradication strategies, while results of the PP analysis should be interpreted as a best-case scenario informing real life clinical practice.”. Please see lines 224-229.
4) Patients with cirrhosis are in need of HCC screening post-SVR. In the study most of the older patients and those with cirrhosis remain in follow-up care. This is a key finding and should be stressed more in the manuscript and be the main conclusion.
We added the following sentence “Importantly, most of the older patients and those with cirrhosis remained in follow-up care, facilitating early detection of possible complications such as HCC. For these patients, early HCC diagnosis remains determinant”. Please see lines 234-236.
5) The terms intention-to-treat vs.per-protocol analysis should be used throughout the manuscript.
We applied the suggested revision and now these terms are appropriately used throughout the manuscript.
6) The authors assume that age, comorbidity and comedication could impact adherence. Please reference.
An appropriate reference has been included (#7). Please see line 58.
7) I propose not to use the first person plural form, this is unusual for scientific manuscripts.
We applied the suggested revision and deleted first person plural form where appropriate in the manuscript.
8) Please provide the total time of observation for the 12 detected HCC cases.
We added the requested information: “Among 184 patients who were retained into care and were alive, 12 (6.5%) developed HCC, which was detected within one year after the EOT”. Please see lines 162-163.
9) It remains unclear whether genotype 3 is pos. or neg risk factor for retention in follow-up care.
Thank for this suggestion and apologize for this mistake. This part was corrected and clarified as follows: “At univariate analysis, variables associated with the risk of being lost to follow-up were: male gender (OR: 2.57, 95%CI: 0.98-7.50, p=0.03), HCV genotype 3 (OR: 4.78, 95%CI: 1.12-18.08, p<0.01), and history of HCC (OR: 7.00, 95%CI: 0.48-98.85, p=0.03)”. Please see lines 169-171.
10) Discussion: please do not use the term "decompensated patients". Better use "patients with decompensated liver cirrhosis.
This correction has been applied, deleting this terms.
11) Discussion: patient with intravenous drug use would most importantly use a follow-up for re-infection assessment. Please discuss.
We added the following sentence: “In these patients the main risk is likely to be continuing HCV transmission if SVR is not obtained and risk behaviours (i.e., intravenous drug use) are still ongoing. Therefore, patients with ongoing intravenous drug use should receive most importantly a follow-up for reinfection assessment”. Please see lines 208-211.
12) Line 189: risk of HCC and progression of liver disease is only a topic for Non-SVR patients, please add this information.
Thank you for your comments. We added the following sentence: “especially if SVR is not achieved”. Please see line 197.

Round 2
Reviewer 2 Report
all raised issues are now met in the revised version.